# *Bondarzewia dickinsii* Against Colitis-Associated Cancer Through the Suppression of the PI3K/AKT/COX-2 Pathway and Inhibition of PGE2 Production in Mice

**DOI:** 10.3390/nu16234048

**Published:** 2024-11-26

**Authors:** Junliang Chen, Shuai Liu, Xin Zhang, Xiaojing Dai, Yu Li, Yonglin Han, Lanzhou Li

**Affiliations:** 1Engineering Research Center of Edible and Medicinal Fungi, Ministry of Education, Jilin Agricultural University, Changchun 130118, China; chenjunliang1026@126.com (J.C.); 20221377@mails.jlau.edu.cn (S.L.); 2021200116@stu.syau.edu.cn (X.Z.); daixiaojing@mail.jlau.edu.cn (X.D.); liyu@jlau.edu.cn (Y.L.); 2Science and Research Center for Edible Fungi of Qingyuan County, Qingyuan 323800, China; 3Tianjin Institute of Industrial Biotechnology, Chinese Academy of Sciences, Tianjin 300308, China; 4National Center of Technology Innovation for Synthetic Biology, Tianjin 300308, China; 5Science Popularization Service Center of Jilin Province, Changchun 130021, China; yonglinhan@163.com

**Keywords:** *Bondarzewia dickinsii*, colitis-associated cancer, prostaglandin E2, PI3K/AKT/COX-2 pathway

## Abstract

Background: *Bondarzewia dickinsii* (BD) is a newly discovered edible mushroom with rich nutritional components. This study presents a thorough analysis of the components of BD, examining its inhibitory effects and the underlying mechanisms by which BD influences colitis-associated cancer (CAC). Methods: AOM/DSS-induced CAC mice (male C57BL/6) were used, and a histopathological analysis, intestinal microbiota assessment, and metabolomics profiling were carried out, as well as an evaluation of relevant proteins and factors, to investigate the CAC-inhibitory effects of BD. Results: BD is rich in nutritional components, including a total sugar content of 37.29% and total protein content of 24.9%. BD significantly diminished colon inflammation, as well as the size and quantity of tumors. In addition, BD modified the diversity of intestinal microbiota and changed the levels of 19 serum metabolites, including arachidonic acid. BD significantly reduced prostaglandin E2 (PGE2) and cyclooxygenase-2 (COX-2) in colon tissue. Furthermore, it was found to inhibit the phosphatidylinositol 3-kinase (PI3K)/protein kinase B (AKT)/COX-2 signaling pathway. Conclusions: In general, BD inhibited the onset and progression of CAC by modulating the composition of intestinal microbiota and metabolite levels, suppressing the PI3K/AKT/COX-2 pathway, and decreasing PGE2 expression. This study provides a significant reference for the development of BD as a dietary supplement and pharmaceutical agent in the treatment of CAC.

## 1. Introduction

Colorectal cancer (CRC) represents a prevalent malignant condition characterized by an increasing incidence and mortality rate [1]. According to the statistical data, CAC is associated with approximately 10% of cancer-related fatalities and newly reported cancer cases worldwide [2]. It is expected that, by 2035, the number of newly reported CRC cases will reach approximately 2.5 million [2]. Colitis-associated cancer (CAC) is a subtype of CRC that is related to inflammatory bowel disease (IBD) [3]. Patients with IBD often develop mutations due to their inflammatory state, which lead to epigenetic changes and genomic instability, ultimately causing CAC [4]. Despite the advances in medical treatments, the chronic inflammation characteristic of IBD heightens the risk of heterotopic hyperplasia and CAC. This heightened risk is shaped by an interplay of genetic factors, environmental influences, and the composition of the microbiome. In comparison to the overall population, individuals diagnosed with IBD demonstrate a 1.4- to 3.0-fold higher probability of developing CAC [5,6]. In summary, the presence of persistent inflammation in the intestinal tissue leads to a substantial risk of the initiation and progression of tumors, rendering the treatment of CRC associated with inflammatory bowel disease a considerable challenge.

In patients with IBD, there is an increase in the expression of inflammation-related genes, including cyclooxygenase-2 (COX-2) and nitric oxide synthase (NOS2) [7]. These factors may have a direct impact on epithelial cells, thereby accelerating the aging process of the colorectal mucosa and heightening the vulnerability to genetic and epigenetic modifications that contribute to the development of cancer [8]. The main product of COX-2 in the inflammatory environment is prostaglandin E2 (PGE2), and its elevated expression is involved in CRC progression [9]. Exposure to PGE2 in human CAC cell lines can lead to dephosphorylation and the parallel nuclear translocation of CREB-regulated transcription co-activator 1 (CRTC1), enhancing its transcriptional activity [10]. Moreover, the stable overexpression of CRTC1 significantly increases the growth of colon tumors [10]. The modulation of inflammatory levels represents a promising treatment for CAC [5].

Microbes in the intestines play an important role in the development of CRC. Changes in the intestinal microbiota can cause specific bacteria to infiltrate the body, which may subsequently alter cellular structure or function within relevant organs, thereby modifying the surrounding organizational environment [11]. *Bacteroides fragilis* is a conditional pathogen, which can be classified into enterotoxigenic and non-enterotoxigenic types [12]. Enterotoxigenic *B. fragilis* was confirmed to facilitate the progress of CAC by activating signal transducer and activator of transcription 3 (STAT3), which in turn stimulates the growth and reproduction of cancer cells [13]. This process triggers the production of interleukin (IL)-6, which activates the STAT3 pathway [14]. *Bifidobacterium* has a great effect on intestinal health maintenance and the treatment of intestinal mucosal diseases [15]. *Bifidobacterium* can change the intestinal microbiota of mice, improve the level of short-chain fatty acids, and increase the expression of intestinal barrier function genes, which significantly alleviates ulcerative colitis in murine models [16]. In addition to its role in alleviating enteritis, *Bifidobacterium* also inhibits the proliferation of CRC cells [17]. The regulation of intestinal microbiota may be a potential therapeutic approach for CAC [18].

In recent years, advancements in treatment methodologies have significantly enhanced the therapeutic outcomes for CRC. However, surgical interventions and pharmacological therapies frequently have pronounced side effects, including impaired digestive function, nausea, loss of appetite, and dysfunctions related to digestion and absorption [19]. As a result, the mortality and prognosis of colon cancer patients have not seen significant improvements. A new, effective, affinity-based, non-toxic, and side-effect-free treatment approach for CRC is crucial.

The utilization of fungi as a therapeutic agent has a profound historical foundation. With the advancements in modern science and technology, research on the pharmacological activity of these fungi has revealed their significant impact on CAC. *Ganoderma lucidum* polysaccharide has been shown to ameliorate the microbiota dysregulation induced by AOM/DSS, enhance the production of short-chain fatty acids, and downregulate the expression of COX-2, IL-1β, and inducible nitric oxide synthase (iNOS) [20]. *Phellinus igniarius* can alleviate colitis by balancing bacteria and metabolites, relieving inflammatory tissue responses, improving antioxidant enzymes’ activity, and increasing the number of short-chain fatty acids [21]. *Hericium erinaceus* can balance the intestinal bacteria, increase the number of short-chain fatty acids, and inhibit GPR41 and GPR43 expression [22]. Previous studies suggest that fungi may hold great potential for treating CAC via the regulation of intestinal microbiota and inflammatory factors.

*Bondarzewia dickinsii* (BD) is a one-year-old large fungus that is soft when fresh and becomes hard after drying. It has a semicircular to fan-shaped appearance; the surface is white to brown when fresh and light yellow to mouse gray when dry. As a newly developed and utilized fungus, the active ingredients and pharmacological activities of BD have not been systematically studied [23].

An AOM/DSS-induced CAC mice model was employed for the screening of novel drugs against CAC and to elucidate their underlying mechanisms [24]. This study systematically analyzed the nutritional composition and content of BD and demonstrated that BD effectively attenuated the phosphatidylinositol 3-kinase (PI3K)/protein kinase B (AKT)/COX-2 pathway in CAC mice, leading to a reduction in PGE2 levels and exhibiting anti-CAC effects. The findings of this study make a significant contribution to the development of dietary nutrition guidelines for BD, as well as the development of therapeutic agents targeting CAC.

## 2. Materials and Methods

### 2.1. Measurement of the BD Components

#### 2.1.1. Main Components

BD fruiting bodies were collected from Qingyuan, Zhejiang, in May 2023 and stored after drying. The main nutritional components of BD fruiting bodies were determined systematically. The phenol–sulfuric acid determination [25], 3,5-dinitrosalicylic acid colorimetric estimation [26], vanillin–glacial acetic acid and perchloric acid colorimetric spectrophotometry [27], aluminum chloride colorimetric method [28], periodate oxidation method [29], petroleum benzine extraction method [30], ashing method [31], Kjeldahl method [32], enzymatic hydrolysis method [33], Liebermann–Burchard reagent [34], and Folin–Ciocalteu method [35] were used to analyze the levels of total sugar, reducing sugar, triterpenoids, flavonoids, mannitol, crude fat, total ash, total protein, crude fiber, sterols, and polyphenols, respectively (Appendix A).

#### 2.1.2. Amino Acid

BD was hydrolyzed with 1:1 HCl (110 °C, 24 h), followed by vacuum-drying. After the drying process, derivatives (ethanol, phenyl isothiocyanate, water, triethylamine in a ratio of 7:1:1:1) were added at room temperature and allowed to react for 30 min. The final product was then analyzed using liquid chromatography [36].

#### 2.1.3. Minerals

An appropriate amount of BD was weighed and added to the Teflon digestion tank; then, 5 mL of nitric acid was added. After the reaction was completed, the lid was sealed and the tank was placed into the microwave digestion instrument. The resolution program was set as follows: temperature was increased to 100 °C and maintained for 3 min; 100 °C was increased to 140 °C and maintained for 3 min; 140 °C was increased to 160 °C and maintained for 3 min; 160 °C was increased to 180 °C and maintained for 3 min; 180 °C was increased to 190 °C and maintained for 15 min. The temperature was then reduced to below 50 °C, and the volume was adjusted to 25 mL using ultrapure water. Subsequently, inductively coupled plasma atomic emission spectrometry (ICP-OES, Perkin Elmer, Waltham, MA, USA) was employed to analyze K, Na, and Ca. Additionally, inductively coupled plasma mass spectrometry (ICP-MS, Thermo Fisher Scientific, Waltham, MA, USA) was utilized for the detection of Zn, Fe, Mn, Cu, Se, Cr, As, Cd, and Pb [37].

#### 2.1.4. Fatty Acids

BD was hydrolyzed with HCl at 80 °C for 1 h. After cooling, 95% ethanol was added; the extract was obtained by drying the ether, adding 2% potassium hydroxide at 45 °C for 30 min, adding 14% boron trifluoride methanol solution at 45 °C for 30 min, and adding n-hexane extraction after cooling. Finally, the fatty acid level was analyzed via gas chromatography (7890A, Agilent, Santa Clara, CA, USA) [38,39].

### 2.2. CAC Mice Model Establishment and BD Treatment Procedure

Thirty-two male C57BL/6 mice (22–24 g, 8 weeks old) were purchased from Beijing HFK Bio-Technology Co., Ltd., (Beijing, China). The mice were placed under specific pathogen-free (SPF) and constant-temperature and -humidity conditions for 12 h. The mice were adapted to the environment for 7 days. The animal study protocol was approved by the Ethics Committee of Jilin Agricultural University (protocol code 2023066002 and date 15 June 2023).

Twenty-four mice were intraperitoneally administered 10 mg/kg of AOM (25843-45-2, Sigma-Aldrich, St. Louis, MO, USA) on the first day. Subsequently, their drinking water was replaced with 2% DSS (9011-18-1, Shanghai Yuanye Bio-Technology Co., Ltd., Shanghai, China) during the second, fifth, and eighth weeks. From the fifth week onward, the mice were randomly assigned to three groups. They received either normal saline (*n* = 8) as a CAC model group or doses of 100 mg/kg or 300 mg/kg BD (Qingyuan, China) daily for a duration of six weeks to serve as the BD treatment group. Another eight mice were administered intraperitoneal injections of normal saline on the first day. Throughout the entire experimental period, they received regular drinking water and were given oral doses of normal saline daily from the fifth to the tenth week; these served as the control (Ctrl) group (Figure 1A). Mice were monitored and weighed on a daily basis. If the weight of the mouse continued to decrease by more than 15%, or the mouse exhibited anorexia for 5 consecutive days, experienced water deprivation for 3 consecutive days, or demonstrated severe debilitation, humane euthanasia was performed. Six hours after the final administration, the mice were subjected to blood collection via the tail vein, followed by euthanasia via carbon dioxide. Subsequently, they were dissected to obtain samples of the colorectum, heart, liver, spleen, kidneys, thymus, and cecal contents. Visible tumors were quantified, and the length and weight of the colon were measured to calculate the colon coefficient.

### 2.3. Histopathological Examination

The polyformaldehyde-fixed tissues of the heart, liver, spleen, kidney, thymus, and colon were embedded in paraffin. Subsequently, the tissues were sliced and dewaxed with xylene, anhydrous ethanol, and 75% ethanol. Sections were stained with hematoxylin and eosin (H&E), followed by dehydration with ethanol and xylene. The sealed sections were analyzed using an optical microscope.

### 2.4. Intestinal Microbiota Analysis

An analysis of intestinal microbiota, conducted using 16S rRNA sequencing, was conducted for the Ctrl mice, CAC model mice, and CAC mice treated with 100 mg/kg BD (*n* = 5). Nucleic acids were extracted from the contents of each cecum, followed by PCR amplification of the V3-V4 region of the bacterial 16S rRNA gene. The sequencing process was carried out at Shanghai Peisenor Biotechnology Co., Ltd., (Shanghai, China).

Utilizing the ASV abundance data, a flower plot was generated, and both alpha diversity indices and the weighted UniFrac distance matrix were calculated. Abundance data from each microbiota group were utilized to conduct an LDA effect size (LEfse) analysis and generate a genus composition heatmap.

### 2.5. Metabolomics Analysis

The plasma from the control group, model group, and group of mice treated with 100 mg/kg BD (*n* = 5) were subjected to centrifugation for 30 min. The upper layer of plasma was then frozen for further analysis. The plasma samples’ proteins were precipitated with a bifid solution mixed with cold methanol/acetonitrile (1:1, *v*/*v*). After the supernatant was completely dried, it was dissolved in a mixture of acetonitrile and water (1:1, *v*/*v*) for a subsequent LC-MS analysis. The score map was generated using the processed data. A differential metabolite heatmap and correlation heatmap were used to analyze metabolites affected by BD. The correlation heatmap of significantly altered metabolites and genus microbiota was used to analyze the correlation between intestinal microbiota and metabolites

### 2.6. Cytokine Assay

The levels of COX-2, PGE2, and IL-6 in the colon were quantified using ELISA kits (Jiangsu Meimian Industrial Co., Ltd., Yancheng, China). The protein concentrations were quantified using a BCA protein assay kit (Invitrogen Co., Ltd., Shanghai, China). The testing procedure was developed in accordance with the provided instructions.

### 2.7. Western Blot

A low-temperature lysis of colon tissue was carried out using an appropriate RIPA lysis buffer. The total protein levels were measured with the BCA protein detection kit. Target proteins were separated using 10% and 12.5% SDS-PAGE gels and subsequently transferred onto PVDF membranes. PVDF membranes were then sealed in a rapid sealing solution at 4 °C for 2 h and incubated with anti-GAPDH, anti-COX-2, anti-AREG, anti-CRTC1, anti-P-AKT, anti-AKT, anti-P-PI3K, anti-PI3K, anti-PKA (from ABclonal, Woburn, MA, USA), and anti-PGE2 (from Absin, Shanghai, China) antibodies at 4 °C for 12 h, before being incubated with an appropriate secondary antibody at 4 °C for 4 h. The imaging process was conducted using the ECL reagent, followed by a semi-quantitative analysis of the resultant images using ImageJ v1.8.0.

### 2.8. Statistical Analysis

SPSS 16.0 software (IBM Corporation, Armonk, NY, USA) was used to analyze the data. The statistical analysis was performed using one-way analysis of variance (ANOVA) followed by Dunn’s test. In the analysis, *p* < 0.05 was considered to be statistically significant.

## 3. Results

### 3.1. Composition of BD

The BD fruiting body comprises 37.29% total sugars, 8.07% reducing sugars, 24.90% total protein, 15.84% mannitol, 11.12% crude fiber, 5.67% total ash content, 3.47% flavonoids, 1.64% crude fat, 0.60% triterpenoids, 0.33% sterols, and 0.008% polyphenols (Table 1). The fruit body of BD contains 16 amino acids; the glutamic (Glu) content comprises 1.38% of the total, alanine (Ala) content comprises 1.10%, and aspartate (Asp) content comprises 1.05%, which is higher than that of other amino acids (Table 2). The BD fruiting body contains the following mineral elements (mg/kg): potassium (K) (2.24 × 10^4^), calcium (Ca) (340.0), iron (Fe) (132.0), sodium (Na) (59.2), zinc (Zn) (55.6), copper (Cu) (21.9), manganese (Mn) (9.84), lead (Pb) (0.519), cadmium (Cd) (0.248), chromium (Cr) (0.162), selenium (Se) (0.0639), and arsenic (As) (0.0633) (Table 3). Eight fatty acids were detected in the fruiting body of BD (Table 4).

### 3.2. BD Markedly Suppresses Tumor Growth in CAC Mice

BD has no effect on the change in body weight (Figure 1B). Compared with the Ctrl mice, the colorectal length of the CAC mice was significantly shortened, and there were multiple instances of hyperplasia protrusion. After the administration of BD, the colorectal length recovered to a certain extent, and the prominent proliferative tissues were significantly reduced (Figure 1C and Figure 2A). The average colorectal lengths of 100 and 300 mg/kg BD-administration mice were 8.5 cm and 8 cm. Compared with the Ctrl mice, the colorectum of the CAC mice was shortened by 18.3% (*p* < 0.001). Compared to the CAC mice, the colorectum of mice administered with 100 mg/kg and 300 mg/kg BD exhibited increases of 11.8% (*p* < 0.001) and 5.3% (*p* < 0.01), respectively (Figure 2B). Compared with the Ctrl mice, the colon coefficients of CAC mice were markedly reduced by 141.3% (*p* < 0.001), and the colon coefficients of 100 and 300 mg/kg BD-administration mice were markedly reduced by 39.7% (*p* < 0.001) and 39.8% (*p* < 0.001) (Figure 2C). According to the H&E staining of mouse colorectal, there is significant tissue proliferation in the colon of CAC mice, and BD treatment significantly inhibits tissue proliferation in the colon of mice. There were no neoplastic cells observed in the colorectal tissue of the control mice; however, a significant number of neoplastic cells were identified in the CAC mice, accompanied by a reduction in goblet cell numbers. The number of neoplasia cells in BD-treated mice was significantly reduced, and a certain number of goblet cells were restored (Figure 2D). Through organ index analysis and HE staining, BD administration was shown to not lead to pharmacological toxicity in the organs of mice (Appendix A).

### 3.3. BD Regulated the Intestinal Microbiota in CAC Mice

Among the 4830 detected Amplicon Sequence Variants (ASVs), 387 were common to all three groups. The distribution of ASVs across the Ctrl, model, and BD groups was as follows: 1870 in the Ctrl group, 1234 in the model group, and 997 in the BD group (Figure 3A). BD induced structural alterations in the microbiota of CAC mice. A beta diversity analysis showed the significant differences in the microbial composition of the Ctrl, model, and BD groups (Figure 3B). It is evident from the α diversity index that BD does not influence the abundance and diversity of microorganisms overall (Appendix A), while BD may influence a particular microorganism. The Linear Discriminant Analysis Effect Size (LEfSe) analysis serves as a robust analytical tool for identifying and interpreting biological characteristics, including taxa, pathways, and genes, within high-dimensional datasets. LEfSe was employed to discern significant differences in the biomarker profiles of intestinal microbiota across various groups. A total of 12 taxa in the BD treatment group were labeled as biomarkers, including *TM7*, *RF39*, *Turicibacter*, *Moraxellaceae*, *Turicibacteraceae,* and *Prevotellaceae*. The model group contains four prominent taxa, *Leptotrichiaceae*, *Anaerotruncus*, *Snerthia*, and *Serratia* (Appendix A). According to the heatmap analysis of the top 50 genera of intestinal microbiota, BD treatment effectively reversed the alterations in intestinal microbiota induced by AOM/DSS. This intervention resulted in an increase in the levels of *Parabacteroides*, *Prevotella*, *Bacteroides*, and *Butyricimonas*, while simultaneously decreasing the levels of *Akkermansia*, *Coprococcus*, *Lactobacillus*, and *Desulfovibrio* (Figure 3C).

### 3.4. BD Altered the Metabolism in CAC Mice

The OPLS-DA score map showed that the metabolite levels of mice in each group changed significantly after the treatments (Figure 4A). A total of 149 distinct metabolites were identified. Among these, 94 exhibited significant differences between the model mice and the Ctrl mice, while 19 showed notable variations between the model mice and the BD-treated mice (Figure 4B). A heatmap analysis of differential metabolites was conducted, demonstrating that the serum levels of 26 metabolites in the BD-treated mice were significantly decreased. These metabolites included arachidonic acid, linolenic acid, cinchonine, fenpropidin, and vindoline (Figure 4C). BD treatment resulted in upregulated levels of 11 metabolites, including L-Asparagine, 5-aminovaleric acid, L-Tryptophan, D-Quinovose, and acetylcarnitine (Figure 4C).

An association analysis was conducted to find the relationship between intestinal microbiota and metabolites. Arachidonic acid was negatively correlated with *Butyricimonas* (*p* < 0.05) and positively correlated with *Turicibacter* (*p* < 0.05). Linolenic acid was negatively correlated with *Butyricimonas* (*p* < 0.05)*, Prevotella* (*p* < 0.05)*, Bacteroides* (*p* < 0.05)*, Butyricimonas* (*p* < 0.05)*,* and [*Eubacterium*] (*p* < 0.05), and positively correlated with *Bifidobacterium* (*p* < 0.05) (Figure 5A).

The main downstream product of arachidonic acid is PGE2. Compared with Ctrl mice, the PGE2 content in the colon of CAC mice was significantly increased, by 31.3% (*p* < 0.001). The PGE2 content in the colon of BD-treated mice was significantly decreased, by 13.7% (*p* < 0.05), compared with CAC mice (Figure 5B). COX-2 serves as a crucial enzyme in the biosynthesis of PGE2. Compared with the Ctrl mice, the COX-2 levels in the colon of CAC mice were significantly increased, by 48.6% (*p* < 0.001). Compared with CAC mice, the COX-2 levels in the colon of mice treated with BD were significantly reduced, by 13.7% (*p* < 0.05) (Figure 5C). Compared with the Ctrl mice, the level of IL-6 in the colon of CAC mice was significantly increased, by 30.5% (*p* < 0.001). Compared with CAC mice, IL-6 levels in the colon of BD-treated mice were significantly decreased, by 13.8% (*p* < 0.05) (Figure 5D).

### 3.5. BD Inhibited CAC by Regulating PI3K/AKT/COX-2 Pathway

To elucidate the relationship between the relieving effect of BD on CAC and the PI3K/AKT/COX-2 pathway, we assessed the relevant proteins. Compared to the Ctrl mice, the levels of P-PI3K, P-AKT, COX-2, and PGES2 in the colon of CAC mice demonstrated significant increases of 50% (*p* < 0.05), 122.9% (*p* < 0.01), 60.2% (*p* < 0.01), and 31.9% (*p* < 0.05), respectively. (Figure 6A–E). The levels of PKA, CRTCI, AREG, and IL-6 exhibited increases of 78.9% (*p* < 0.01), 109.8% (*p* < 0.05), 134.7% (*p* < 0.01), and 51.6% (*p* < 0.01), respectively (Figure 6A,F–I). Compared to CAC mice, the levels of P-PI3K, P-AKT, COX-2, and PGES2 in the colon of mice treated with 100 mg/kg BD were reduced by 66.6% (*p* < 0.01), 75.8% (*p* < 0.001), 47.7% (*p* < 0.01), and 47.5% (*p* < 0.01), respectively (Figure 6A–E). The levels of PKA, CRTCI, AREG, and IL-6 exhibited increases of 56.6% (*p* < 0.01), 55.3% (*p* < 0.05), 78% (*p* < 0.001), and 49.1% (*p* < 0.01), respectively (Figure 6A,F–I). Compared to CAC mice, the levels of P-PI3K, P-AKT, COX-2, and PGES2 in the colon of mice treated with 300 mg/kg BD were reduced by 56.4% (*p* < 0.01), 61.1% (*p* < 0.01), 47.8% (*p* < 0.05), and 46.3% (*p* < 0.05), respectively (Figure 6A–E). The levels of PKA, CRTCI, AREG, and IL-6 exhibited increases of 25.6%, 49.8% (*p* < 0.05), 48.8% (*p* < 0.05), and 27.4% (*p* < 0.01), respectively (Figure 6A,F–I).

## 4. Discussion

In this study, the composition of BD was systematically identified. The total sugar content was established to be 37.29%, while the total protein content was measured at 24.9% in BD. Polysaccharides extracted from mushrooms were found to have many beneficial activities, including antioxidant properties and anti-tumor effects, as well as playing a role in the regulation of blood sugar and lipid levels, the prevention of osteoporosis, the prevention of alcohol-induced liver damage, and the enhancement of immune function [40,41,42,43]. The mushroom protein is a crucial physiological active substance involved in immune regulation, oxidative metabolism, tissue repair, and other biological processes [44,45,46,47]. The contents of the other nutrients were as follows: mannitol (15.84%) > crude fiber (11.12%) > total ash (5.67%) > total flavonoids (3.47%) > crude fat (1.64%) > total triterpenoids (0.60%) > total sterols (0.33%) > total phenols (0.08%). Amino acids, which can affect the metabolism of prostaglandins, are often closely related to the anti-inflammatory effects of fungi. BD is rich in amino acids, including Asp, Glu, and Thr. The antitumor efficacy of a novel arylphosphamide prodrug, derived from Asp and Glu, was demonstrated [48]. Thr has significant immunomodulatory effects, boosting both innate (phagocytic) and adaptive (immune response) immunity [49]. In terms of composition, BD is a nutrient-rich substance with significant medicinal potential.

The protective effects of BD against CAC were then investigated using an AOM/DSS-induced CAC mouse model. BD effectively mitigated tumor progression, reduced PGE2 levels, inhibited the activation of the PI3K/AKT/COX-2 pathway, and modulated the expression of downstream proteins and cell factors by regulating specific intestinal microorganisms and metabolites.

Changes in the intestinal microbiota play a crucial role in the occurrence and development of tumors. *Akkermansia* can increase host susceptibility to AOM/DSS-induced CRC by stimulating the production of pro-inflammatory cytokines such as IL-6, promoting inflammatory cell infiltration, reducing the number of goblet cells, and promoting the proliferation of intestinal epithelial cells [50,51]. The genera *Butyricimonas* and *Bacteroides* play a critical role in the production of short-chain fatty acids, which are essential metabolites that contribute to the maintenance of immune system stability and intestinal homeostasis [52]. The short-chain fatty acids present in the intestine have the ability to regulate DNA methylation within the rectal mucosa, thereby influencing gene stability and subsequently modifying the probability of intestinal carcinogenesis [53]. In this study, BD treatment resulted in an increase in *Prevotella* and *Bacteroides*, while *Akkermansia* exhibited a decrease. These results suggest that the inhibitory effect of BD on CAC may be attributed to its modulation of specific intestinal microbiota and alteration of intestinal inflammatory responses.

Metabolites play a pivotal role in tumor development. Following BD treatment, the levels of arachidonic acid and linolenic acid in the plasma of CAC mice decreased significantly. In the correlation analysis of intestinal microbiota and metabolites, the levels of arachidonic acid have a negative correlation with *Butyricimonas*, whereas they exhibit a positive correlation with *Turicibacter*. Linolenic acid exhibited a negative correlation with *Butyricimonas*, *Prevotella*, *Bacteroides*, and *Eubacterium*, while demonstrating a positive correlation with *Bifidobacterium*. Arachidonic acid can enzymatically synthesize PGE2 via COX-2 [54], an enzyme frequently overexpressed in various cancer types, exerting pleiotropic and multifaceted effects on carcinogenesis initiation or promotion, as well as contributing to chemotherapy and radiotherapy resistance in cancer cells [55,56,57]. PGE2 can promote the development of mouse CRC stem cells, thereby promoting the progression of CRC by enhancing the transcriptional activity of CRTC1 [10]. In numerous malignancies, the activation of the PI3K/AKT signaling pathway upregulates COX-2 expression [58,59,60]. BD treatment also significantly reduced the levels of COX-2, PGE2, and IL-6 in the colon of CAC mice, which suggests that the inhibitory effect of BD on CAC may be related to the metabolic regulation of PGE2.

The PI3K/AKT pathway plays a crucial role in the growth and spread of CRC cells, and increases COX-2 production, thereby enhancing cell aggressiveness [61]. In CRC, COX-2 overproduction is associated with increased PGES2 production, which contributes to inflammation in intestinal tissue [62]. PGE2 increases cyclic adenosine monophosphate (cAMP) and intracellular Ca^2+^ levels through EP1 and EP2 receptors, activating PKA, which mediates CRTC1 signaling by promoting the transcriptional enhancement of tumor-promoting genes in human colon cancer cells through CREB and AP1 [10]. This markedly elevates the mRNA levels of NR4A2, COX2, and the regulatory proteins AREG and IL-6, thereby facilitating the development of sporadic or colitis-associated colon cancer [10]. Elevated levels of IL-6 have been detected in various tumors, including those associated with gastrointestinal cancers [63]. Invasive immune cells in tumors promote IL-6 production [64]. IL-6 activates Signal Transducer and Activator of Transcription 3 (STAT3), thereby facilitating tumor initiation and growth [65]. AREG, an epidermal growth factor receptor (EGFR) ligand, is upregulated in colorectal cancer (CRC) tissue and is associated with tumor invasion depth, nerve infiltration, and liver metastasis [66]. BD treatment effectively suppresses the phosphorylation of proteins in the PI3K/AKT pathway, resulting in a significant reduction in COX-2 and PGES2 protein expression levels, which may lead to a notable decrease in PGE2 levels. These alterations subsequently result in a decline in PKA, AREG, and IL-6 expression levels. These findings may elucidate the mechanism through which CAC is inhibited by BD.

There are some limitations to this study. The specific constituents responsible for the inhibition of CAC activity in mushrooms remain unidentified. Furthermore, the inhibitory effect of BD against CAC was only confirmed in AOM/DSS mice; the potential of BD to inhibit the levels of PGE2 and exert anti-CAC activity in patients with CAC remains to be investigated. The preclinical research findings of this study substantiate the anti-CAC activity of BD. Compared to clinical settings, preclinical studies offer greater experimental control, shorter research cycles, and a reduced potential for adverse effects. In future studies, the active ingredients in BD could be isolated and comprehensive research could be conducted on the efficacy of BD and its active components in treatments or adjunctive therapy for CAC, while also promoting clinical investigations.

## 5. Conclusions

In this study, BD was found to inhibit the onset and progression of CAC by changing the composition of intestinal microbiota and metabolite levels. Additionally, it suppressed the PI3K/AKT/COX-2 signaling pathway and reduced the expression of PGE2, IL-6, and AREG. This study serves as a valuable reference for the development of BD as a therapeutic agent in the treatment of CAC.

## Figures and Tables

**Figure 1 nutrients-16-04048-f001:**
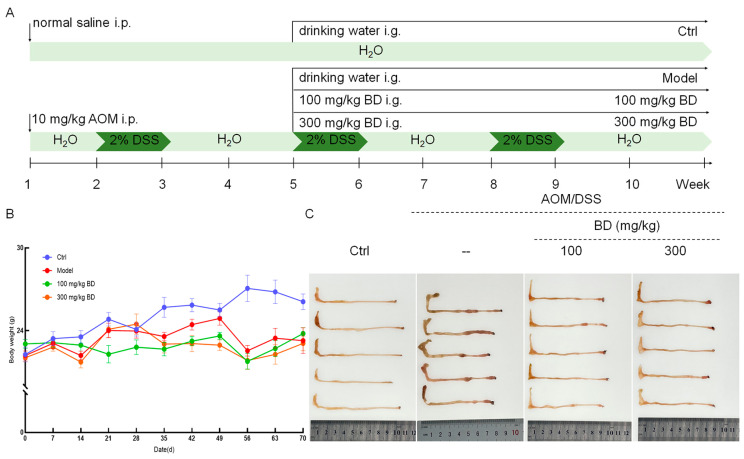
Effect of BD on AOM/DSS-induced CAC mice. (**A**) Flow chart of experimental protocol. (**B**) Bodyweight changes in mice (*n* = 8). (**C**) BD inhibits tumor development in CAC mice (*n* = 5).

**Figure 2 nutrients-16-04048-f002:**
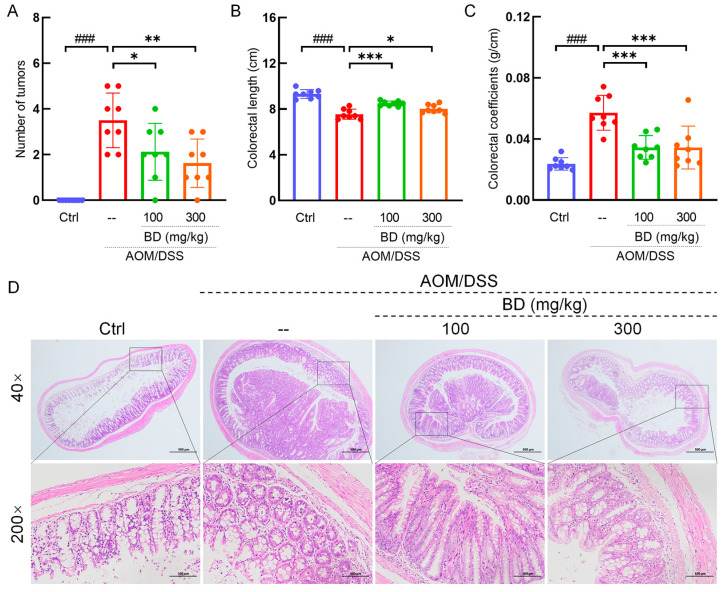
BD alleviates colorectal status in CAC mice. (**A**) The number of tumors. (**B**) The length of the colorectal. (**C**) Colon coefficient. (**D**) Histopathological observation of colorectal tumors in CAC mice (*n* = 3) (40× scale bar: 500 μm; 200× scale bar: 100 μm). ### *p* < 0.001 vs. Ctrl group; * *p* < 0.05, ** *p* < 0.01, and *** *p* < 0.001 vs. model group.

**Figure 3 nutrients-16-04048-f003:**
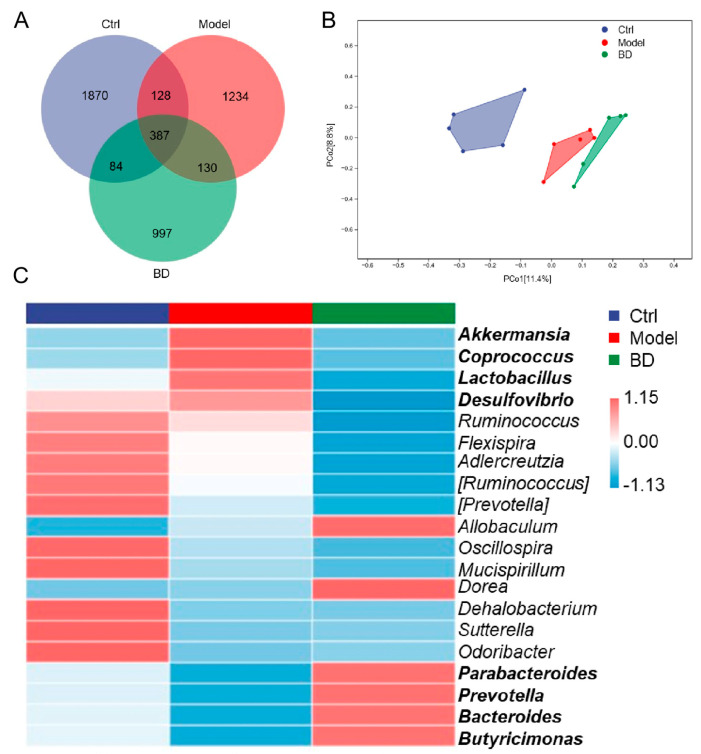
The impact of BD on the intestinal microbiota in CAC mice. (*n* = 4) (**A**) Venn diagram. (**B**) PCoA based on weighted UniFrac distances. (**C**) Heatmap illustrating the composition of the top 20 most advantageous genera.

**Figure 4 nutrients-16-04048-f004:**
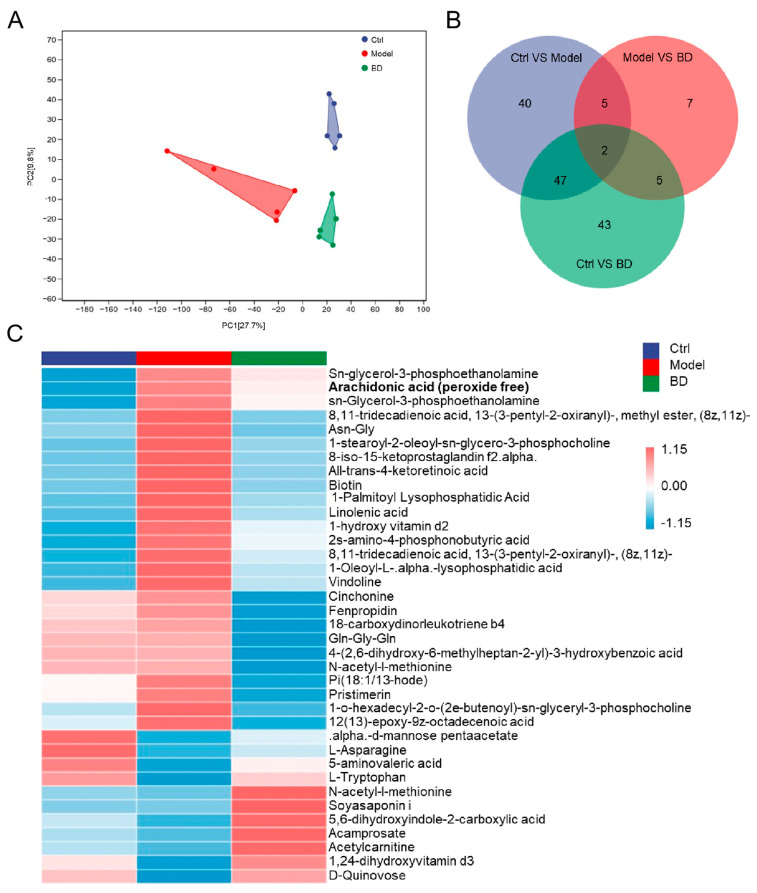
BD modulates serum metabolite concentrations and colorectal cytokine levels in CAC mice. (**A**) OPLS-DA score plot. (**B**) Venn diagram. (**C**) Heatmap of 37 significantly altered metabolites in CAC mice.

**Figure 5 nutrients-16-04048-f005:**
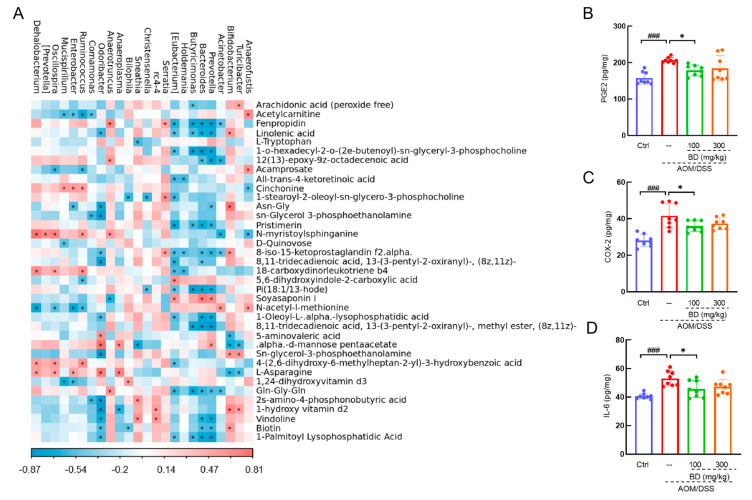
(**A**) The associated heatmap of significantly altered metabolites and genus microbiota. (**B**) PGE2, (**C**) COX-2, (**D**) IL-6 in CAC mice colon. ### *p* < 0.001 vs. Ctrl group; * *p* < 0.05 vs. model group.

**Figure 6 nutrients-16-04048-f006:**
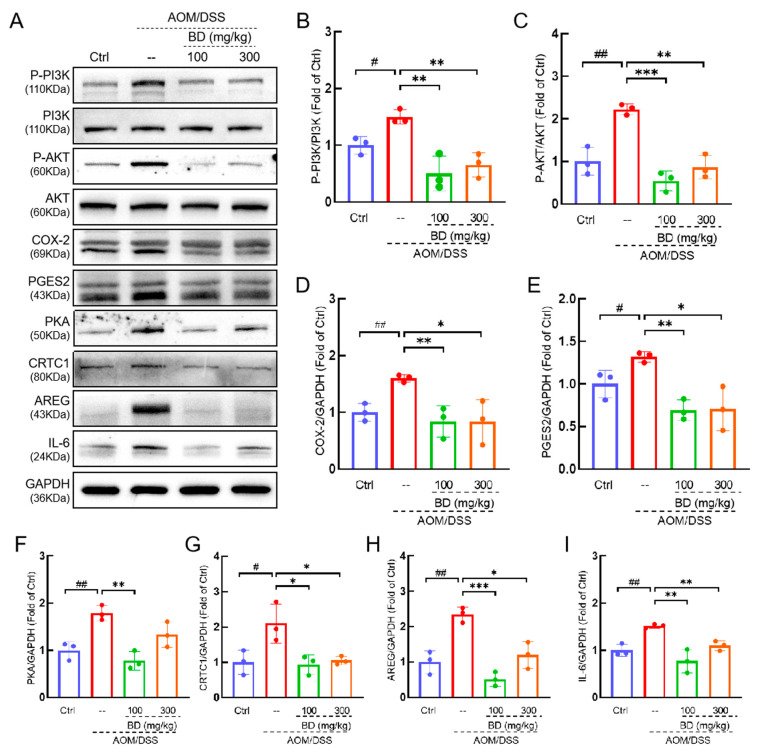
BD regulated protein expression in the colorectal tissue of CAC mice. (**A**) BD reduced the phosphorylation of PI3K and AKT and decreased the levels of COX-2, PGE2 and other proteins. Quantification of (**B**) P-PI3K, (**C**) P-AKT, (**D**) COX-2, (**E**) PGES2, (**F**) PKA, (**G**) CRTC1, (**H**) AREG, and (**I**) IL-6. # *p* < 0.05, ## *p* < 0.01 vs. Ctrl group; * *p* < 0.05, ** *p* < 0.01, and *** *p* < 0.001 vs. model group.

**Table 1 nutrients-16-04048-t001:** Analysis of the regular nutritive components of *Bondarzewia dickinsii*.

Compounds	Contents (%)	Compounds	Contents (%)	Compounds	Contents (%)
Total sugar	37.29	Reducing sugar	8.07	Triterpenoids	0.60
Flavonoids	3.47	Mannitol	15.84	Crude fat	1.64
Total ash	5.67	Total protein	24.90	Crude fiber	11.12
Sterols	0.33	Polyphenols	0.08		

**Table 2 nutrients-16-04048-t002:** Analysis of the amino acid composition of *Bondarzewia dickinsii*.

Compounds	Contents (%)	Compounds	Contents (%)	Compounds	Contents (%)
Glutamic (Glu)	1.38	Aspartic (Asp)	1.05	Leucine (Leu)	1.10
Lysine (Lys)	0.64	Arginine (Arg)	0.75	Alanine (Ala)	0.96
Valine (Val)	0.97	Threonine (Thr)	0.59	Glycine (Gly)	0.50
Serine (Ser)	0.60	Proline (Pro)	0.77	Phenylalanine (Phe)	0.67
Isoleucine (Iso)	0.80	Histidine (His)	0.24		
Methionine (Met)	0.19	Tyrosine (Tyr)	0.36		

**Table 3 nutrients-16-04048-t003:** Analysis of the minerals and heavy metals composition of *Bondarzewia dickinsii*.

Compounds	Contents (mg/kg)	Compounds	Contents (mg/kg)
Manganese (Mn)	9.84	Arsenic (As)	0.0633
Zinc (Zn)	55.6	Kalium (K)	2.24 × 10^4^
Ferrum (Fe)	132	Calcium (Ca)	340
Cadmium (Cd)	0.248	Lead (Pb)	0.519
Cuprum (Cu)	21.9	Chromium (Cr)	0.162
Natrium (Na)	59.2	Selenium (Se)	0.0639

**Table 4 nutrients-16-04048-t004:** Analysis of fatty acids composition of *Bondarzewia dickinsii*.

Compounds	Contents (g/100 g)	Compounds	Contents (g/100 g)	Compounds	Contents (g/100 g)
Butryic acid (C4:0)	ND	Heptadecanoic acid (C17:0)	0.92 × 10^−2^	Docosanoic acid (C22:0)	ND
Caprylic acid (C6:0)	ND	Heptadecenoic acid (C17:1)	ND	Eicosatrienoic acid (C20:3n6)	ND
Octoic acid (C8:0)	ND	Stearic acid (C18:0)	38.39 × 10^−2^	Eicosatrienoic acid (C20:3n3)	ND
Capric acid (C10:0)	ND	Trans-oleic acid (C18:1n9t)	ND	Erucic acid (C22:1n9)	ND
Undecanoic acid (C11:0)	ND	Oleic acid (C18:1n9c)	36.31 × 10^−2^	Arachidonic acid (C20:4n6)	ND
Lauric acid (C12:0)	ND	trans-Linoleic acid (C18:2n6t)	ND	Tricosanoic acid (C23:0)	ND
Tridecanoic acid (C13:0)	ND	Linoleic acid (C18:2n6c)	109.35 × 10^−2^	cis-13,16-Docosadienoic acid (C22:2)	ND
Myristic acid (C14:0)	ND	Arachidic acid (C20:0)	ND	Eicosapentaenoic acid (C20:5n3)	ND
Myristoleic acid (C14:1)	ND	γ-linolenic acid (C18:3n6)	ND	Tetracosanoic acid (C24:0)	0.89 × 10^−2^
Pentadecanoic acid (C15:0)	1.06 × 10^−2^	α-Linolenic acid (C18:3n3)	1.12 × 10^−2^	Nervonic acid (C24:1)	ND
Pentadecenoic acid (C15:1)	ND	Eicosenoic acid (C20:1)	ND	Docosahexaenoic acid (C22:6n3)	ND
Hexadecanoic acid (C16:0)	23.58 × 10^−2^	Heneicosanoic acid (C21:0)	ND		
Palmitoleic acid (C16:1)	ND	Eicosadienoic acid (C20:2)	ND		

## Data Availability

The original contributions presented in this study are included in the article/Appendix A. Further inquiries can be directed to the corresponding author.

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
