# Peer review of "Bondarzewia dickinsii Against Colitis-Associated Cancer Through the Suppression of the PI3K/AKT/COX-2 Pathway and Inhibition of PGE2 Production in Mice"

_nutrients, 2024, doi:10.3390/nu16234048_

Round 1
Reviewer 1 Report
Comments and Suggestions for Authors
The present paper evaluates the implications of Bondarzewia dickinsii species against colitis-associated cancer by suppressing the PI3K/AKT/COX-2 pathway and attenuating PGE2 production. The topic is relevant and challenging with potentially major implications if the results are confirmed and clinically proven. However, there are still a number of questions and recommendations that need to be answered before it can be considered for publication:
1) The aim of the paper should be improved in terms of describing the contribution to the field under analysis and the elements of scientific novelty presented because the authors only presented in the last paragraph of the introduction what they did in the study
2) In 2.1.1 each analytical method should be correlated in a figure or text form with the type of analyte evaluated
3) How were the analytical methods validated? I refer here to the testing of validation parameters (accuracy, precision, etc.).
4) The limitations paragraph should be better detailed as limitations should be clearly pointed out, not just in general.
5) It would be very interesting if it would also be presented how the authors see that their research could be pursued and what is the benefit of their preclinical research to be chosen over other research for pretesting in a clinical setting
Author Response
Dear Reviewer, We quite appreciate your favorite consideration and insightful comments concerning our manuscript. The comments are very valuable and helpful for improving the quality and readability of our paper, as well as the important guiding significance to our future researches. We have studied the comments carefully and have revised the manuscript exactly. We hope this revision can meet with approval. Revised portion are marked in red in the revised manuscript and the main revisions are as follows:
The present paper evaluates the implications of Bondarzewia dickinsii species against colitis-associated cancer by suppressing the PI3K/AKT/COX-2 pathway and attenuating PGE2 production. The topic is relevant and challenging with potentially major implications if the results are confirmed and clinically proven. However, there are still a number of questions and recommendations that need to be answered before it can be considered for publication:
1) The aim of the paper should be improved in terms of describing the contribution to the field under analysis and the elements of scientific novelty presented because the authors only presented in the last paragraph of the introduction what they did in the study
Reply: Thank you for your comment. Corresponding modifications have been made in Abstract and Introduction.
2) In 2.1.1 each analytical method should be correlated in a figure or text form with the type of analyte evaluated
Reply: Thank you for your comment. The correspondence between the analysis methods and the components has been provided in Supplementary Table s1.
3) How were the analytical methods validated? I refer here to the testing of validation parameters (accuracy, precision, etc.).
Reply: The methods for determining the components in this study were based on mature research methods that have been published. Reference citations have been made in the corresponding locations. Therefore, method validation is not necessary
4) The limitations paragraph should be better detailed as limitations should be clearly pointed out, not just in general.
Reply: Thank you for your suggestion. The limitation has been revised accordingly.
5) It would be very interesting if it would also be presented how the authors see that their research could be pursued and what is the benefit of their preclinical research to be chosen over other research for pretesting in a clinical setting
Reply: Thank you for your suggestion. It has been provided in Discussion accordingly.
Reviewer 2 Report
Comments and Suggestions for Authors
The manuscript entitled "Bondarzewia dickinsii against colitis-associated cancer by suppressing the PI3K/AKT/COX-2 pathway and attenuating PGE2 3 production" aimed to evaluate the effects of a newly discovered edible mushroom on colon cancer in mice as well as the nutritional characteristics of the mushroom.
The major issue of this manuscript is the low number of animals used in the study. Therefore, the title is too pretentious and the work can be considered as a "communication".
Other comments:
"According to statistical data, it is projected that there will be 555,477 new cases of CRC and 286,162 fatalities in China in the year 2020 [2]." The authors project data for the 2020 and we are in the 2024 (?!) - line 35. Kindly provide newest references
2.8 Statistical analysis - type of tests used must be provided
Figure 3, 4 and 5 - unreadable and better resolution/larger fonts should be used
The discussion is too meagre
References are not listed according to the journal's instruction, and double numbering is present.
Author Response
Dear Reviewer,
We quite appreciate your favorite consideration and insightful comments concerning our manuscript. The comments are very valuable and helpful for improving the quality and readability of our paper, as well as the important guiding significance to our future researches. We have studied the comments carefully and have revised the manuscript exactly. We hope this revision can meet with approval. Revised portion are marked in red in the revised manuscript and the main revisions are as follows:
The manuscript entitled "Bondarzewia dickinsii against colitis-associated cancer by suppressing the PI3K/AKT/COX-2 pathway and attenuating PGE2 3 production" aimed to evaluate the effects of a newly discovered edible mushroom on colon cancer in mice as well as the nutritional characteristics of the mushroom.
The major issue of this manuscript is the low number of animals used in the study. Therefore, the title is too pretentious and the work can be considered as a "communication".
Reply: The title has been revised and emphasized that the study was conducted in mice. In future research on the anti-CAC activity of Bondarzewia dickinsii active ingredients, the number of animals in each group will be increased. Thank you for your comment.
Other comments:
"According to statistical data, it is projected that there will be 555,477 new cases of CRC and 286,162 fatalities in China in the year 2020 [2]." The authors project data for the 2020 and we are in the 2024 (?!) - line 35. Kindly provide newest references
Reply: Thank you for your comment. It has been revised accordingly.
2.8 Statistical analysis - type of tests used must be provided
Reply: Thank you for your comment. The type of tests has been provided.
Figure 3, 4 and 5 - unreadable and better resolution/larger fonts should be used
Reply: Thank you for your comment. Figure 3, 4 and 5 has been has been reorganized and exported to improve figure readability. Part of the data has been provided in the form of table.
The discussion is too meagre
Reply: The discussion has been rewritten to provide more information.
References are not listed according to the journal's instruction, and double numbering is present.
Reply: The style of reference has been modified.
Reviewer 3 Report
Comments and Suggestions for Authors
submitted a paper entitled Bondarzewia dickinsii against colitis-associated cancer by suppressing the PI3K/AKT/COX-2 pathway and attenuating PGE2 3 production. Bondarzewia dickinsii is a newly discovered edible mushroom with rich nutritional components. The authors performed an analysis of these components in the context of colitis-associated cancer. They tried to prove that Bondarzewia dickinsii can be considered a therapeutic agent in the treatment of this kind of cancer. More specifically, according to the authors, it inhibited the onset and progression by modulating the composition of intestinal microbiota and metabolite levels. It is an original paper that can be motivation to further study this issue. The paper is well-prepared. The authors should carefully check the text and correct errors according to the Guides for authors. They should also eliminate editorial errors, such as Line 275 – double dot; Double numbering of references, etc. Fig. 3-5 contains too much data and is not readable (fonts also too small) – maybe authors should include some data from this fig. in the supplementary data?
Author Response
Dear Reviewer,
We quite appreciate your favorite consideration and insightful comments concerning our manuscript. The comments are very valuable and helpful for improving the quality and readability of our paper, as well as the important guiding significance to our future researches. We have studied the comments carefully and have revised the manuscript exactly. We hope this revision can meet with approval. Revised portion are marked in red in the revised manuscript and the main revisions are as follows:
Submitted a paper entitled Bondarzewia dickinsii against colitis-associated cancer by suppressing the PI3K/AKT/COX-2 pathway and attenuating PGE2 3 production. Bondarzewia dickinsii is a newly discovered edible mushroom with rich nutritional components. The authors performed an analysis of these components in the context of colitis-associated cancer. They tried to prove that Bondarzewia dickinsii can be considered a therapeutic agent in the treatment of this kind of cancer. More specifically, according to the authors, it inhibited the onset and progression by modulating the composition of intestinal microbiota and metabolite levels. It is an original paper that can be motivation to further study this issue. The paper is well-prepared. The authors should carefully check the text and correct errors according to the Guides for authors. They should also eliminate editorial errors, such as Line 275 – double dot; Double numbering of references, etc. Fig. 3-5 contains too much data and is not readable (fonts also too small) – maybe authors should include some data from this fig. in the supplementary data?
Reply: Thank you very much for your comments.
The text has been checked and revised carefully.
The style of reference has been modified.
Figure 3, 4 and 5 has been has been reorganized and exported to improve figure readability. Part of the data has been provided in the form of table.
Round 2
Reviewer 1 Report
Comments and Suggestions for Authors
The authors have significantly improved the manuscript based on the suggestions received.